# Topside Ionospheric Tomography Exclusively Based on LEO POD GPS Carrier Phases: Application to Autonomous LEO DCB Estimation

Manuel Hernández-Pajares [1,*], Germán Olivares-Pulido [1], M. Mainul Hoque [2], Fabricio S. Prol [2], Liangliang Yuan [2], Riccardo Notarpietro [3] and Victoria Graffigna [1]

1    UPC-IonSAT/IEEC-UPC, Mod.C3 Campus Nord UPC, C/Jordi Girona 1-3, E-08034 Barcelona, Spain
2    German Aerospace Center (DLR), 17235 Neustrelitz, Germany
3    EUMETSAT, 64295 Darmstadt, Germany
*    Correspondence: manuel.hernandez@upc.edu; Tel.: +34-93-4016019

**Abstract:** This paper presents a novel technique to estimate DCBs from GPS transmitters and receivers on-board Low Earth Orbit (LEO) satellites. The technique consists of obtaining the DCBs as residuals from the difference between the ionospheric combination of the code and the associated ionospheric delay. The ionospheric delay is computed with TOMION, a background-model-free ionospheric tomographic technique based on dual-frequency GPS carrier phase data only, and solved with a Kalman filter. Thus, DCBs are also estimated epoch-wise from the LEO Precise Orbit Determination (POD) GPS receiver as a secondary product. The results for GPS satellite DCBs, obtained exclusively from the three MetOp LEO POD GPS receivers over four consecutive weeks, are in full agreement (i.e., at the level of a few tenths of ns) with those reported independently with other techniques from hundreds of ground-based receivers exclusively, by JPL and CODE analysis centers.

**Keywords:** ionospheric tomography; Global Navigation Satellite Systems; Low Earth Orbiting satellites; differential code biases

## 1. Introduction

The inter-frequency differential code bias (DCB) is usually one of the most important error sources that affect the accuracy of slant total electron content (TEC) estimation from ground- as well as space-based measurements of the Global Navigation Satellite System (GNSS). They can be understood as hardware delays caused by the digital signal processing components, antenna and cables, when signals at the satellite and receiver are encoded. For example, ref. [1] quantified timing delays between the different legacy and modernized GPS and Galileo signals broadcast on $L_1$ and their dependencies on factors such as user receiver filter bandwidth, filter transfer function and delay locked loop (DLL) correlator spacing; ref. [2] provided a comprehensive overview of pseudorange biases and their dependency on receiver front-end bandwidth and correlator design, and [3] showed that the long-term variations of current DCB products may vary significantly.

The most common DCB calibration methods can provide high accuracy in single-frequency precise point positioning (ref. [4]), it being common to estimate them in ground-based stations (ref. [5]) and to consider GNSS satellite and receiver DCBs as constant during a single day (refs. [6,7]).

Over the last two decades, however, significant research concerning the Global Positioning System (GPS) DCB has been conducted. For example, ref. [8] analyzed the stability of the GPS instrumental biases; ref. [9] found that the day-to-day and annual variation in the estimated GPS DCB is related to the ionospheric variability; and [10] concluded that DCB variability is attributed to the GPS satellite replacements with different satellite types and the zero-mean condition imposed on all satellite DCBs. Furthermore, refs. [11,12]

found intraday variabilities with clear correlations in temperature. Estimate methods for DCBs have also been developed for on-board Low Earth Orbit (LEO) satellites. In this regard, ref. [13] showed that the STEC estimate might be enhanced if the temperature dependency of DCB estimation is considered as well. Ref. [14] estimated the COSMIC GPS DCB receivers in order to perform plasmaspheric observations from LEO satellites with GNSS data. Moreover, ref. [15] developed a method whereby GNSS observations from standalone LEO satellites (i.e., with no ground-based GNSS data) can estimate GPS satellite DCBs.

Various methods for DCB estimation in LEO satellites were proposed in the past decade. For example, least squares-based estimation using a Fengyun-3C satellite was developed by [16]. A zero method was proposed and tested using long-term observations of the Challenging Minisatellite Payload (CHAMP) and Gravity Recovery and Climate Experiment (GRACE) by [17]. GPS P1-P2 DCBs and vertical TEC (VTEC) above orbit height were estimated with JASON-2 and precise orbit determination (POD) observations epoch by epoch by [18]. Ref. [19] estimated the GPS and Beidou satellite DCBs as well as receiver DCB onboard Fengyun-3C using the same method as [18]. Ref. [15] proposed an approach to estimate the GPS satellite DCB based on the ionospheric spherical symmetry assumption for CHAMP and Constellation Observing System for Meteorology, Ionosphere and Climate (COSMIC) observations. It was found that the GPS satellite DCB estimates based on CHAMP were much better than those based on COSMIC. Refs. [20,21] proposed an inequality constrained least squares method to estimate the COSMIC receiver DCBs with LEO satellites.

In this research, we study the capability of performing topside ionospheric tomography based exclusively on dual-frequency carrier phase LEO POD GPS data and validated in terms of the precision and accuracy of the GPS transmitter DCBs. The advantages of using tomography include the consideration of several ionospheric layers, instead of the common use of unrealistic mapping functions to convert VTEC into the slant direction.

A few studies have already developed tomography approaches to obtain topside electron density representations. For example, ref. [22] presented initial results of the sounding of the topside ionosphere and plasmasphere based on GPS measurements from CHAMP. Ref. [23] introduced a new mathematical approach to imaging the electron density distribution in the high regions of the topside ionosphere and the plasmasphere using GPS measurements from LEO satellites. A data assimilative method based on 3D Var for the sounding of the ionosphere and plasmasphere using COSMIC-GPS measurements was introduced in [24]. Using Jason-1 plasmaspheric total electron content (TEC) measurements, ref. [25] developed a tomographic technique for the reconstruction of the plasmaspheric density. COSMIC satellite data have also been used for a tomographic method that estimates the plasmasphere [26]. However, the common approach is to calibrate the DCB unknowns before the tomography inversion. In [27], the DCBs are estimated simultaneously with the electron densities, regarding background values and model respectively, from input GPS ground- and LEO-based (radio occultation) measurements. In this work, the GPS satellite DCB and on-board LEO DCBs are directly retrieved (without background values) right after the 3D electron density distributions are estimated. This estimation is exclusively based on dual-frequency GPS carrier phase measurements gathered from the LEO zenith antennas (i.e., without incorporating radio occultation measurements) and without any electron density background model.

In this regard we processed 1Hz dual-frequency GPS measurements of the POD receivers of the three MetOp LEO satellites, metb (M01), meta (M02) and metc (M03), provided by the European Organization for the Exploitation of Meteorological Satellites (EUMETSAT) from days 173 to 255, 2020.

## 2. TOMION in a Nutshell

TOMION is a data-driven ionospheric model mostly developed by the first author of this work during more than 25 years (refs. [28,29]), in collaboration with Raul Orus-

Perez and David Roma-Dollase for the kriging interpolation and multi-GNSS extensions (see [30,31]). TOMION allows a tomographic estimation by a Kalman filter of the number density of free ionospheric electrons from GPS carrier phase dual-frequency data only and without any background model. Then, full ionospheric maps are obtained via a kriging-based interpolation [30] for the vertically integrated electron density (the vertical total electron content, VTEC). Recently, TOMION has been modified with the capability of processing multi-GNSS measurements (ref. [31], pp. 19–22).

The tomographic model assumes that the ionospheric electron content is distributed in voxels throughout the ionosphere and thus the ionospheric, also known as (hereinafter, a.k.a.) geometry-free (GF), combination ($L_{GF} \equiv L_I$) of two GNSS carrier phases in length units, $L_1$ and $L_2$ with frequencies $f_1 > f_2$, can be expressed through the fundamental equation Equation (1) solved in TOMION,

$$L_{GF,j} \equiv L_1 - L_2 = I + B_{GF,j} = \alpha \cdot S + B_{GF,j} \simeq \alpha \cdot \sum_j N_{e,j} \cdot l_j + B_{GF,j} \qquad (1)$$

where $\alpha = 40.3 \left( \frac{1}{f_2^2} - \frac{1}{f_1^2} \right)$ in International System of Units (IS) ($\alpha \simeq 0.105$ m/TECU for $L_1$ and $L_2$ GPS frequencies, see, for example, [32]), $S$ represents the Slant Total Electron Content (STEC), the index $j$ runs over the number of illuminated voxels and $l_j$ is the segment of the straight line between satellite and receiver that pierces the $j^{th}$ voxel. $N_e$ is the ionospheric electron density and $B_{GF,j}$ represents the corresponding carrier phase ambiguity, both solved as an approximate random walk and random variable (constant), respectively, in a Kalman filter. The electron density random process is estimated with an update of the covariance matrix, with a process noise of $9 \cdot 10^{-8}$ (meters of $L_1 - L_2$ delay/km)$^2$/h. This optimal approximation has been established empirically, after 25 years of daily processing for contributing to IGS with the UPC-IonSAT global ionospheric maps (GIMs), among different tomographic experiments. The initial observation covariances are assumed to be diagonal, with an a priori standard deviation per $L_1 - L_2$ observation equation of 0.05 m.

TOMION is the software used in the generation of UPC-IonSAT GIMs of VTEC for IGS, such as the UQRG one, one of the best, RMS-wise, GIMs in IGS (ref. [33]. UQRG GIM has been assessed with other GIMs and it provides RMS values of 2 TECU [34,35]).

The UQRG GIM produced by TOMION is, for instance, able to detect realistic features of the polar ionosphere as well (ref. [36]) and to provide a realistic and sensitive storm index (ref. [37]). The tomography performed by TOMION is able to ingest different geometries and types of input measurements (ref. [38]), in agreement with independent measurements and models (refs. [39,40]).

## 3. Methodology for DCB Retrieval

The method introduced in this work computes DCBs as the difference between the Geometry-Free combination of the pseudodistance (a.k.a. code or pseudorange) and the associated ionospheric delay estimated with TOMION, a tomographic model without using background electron density and without using GNSS pseudorange measurements (significantly affected by thermal noise and multipath, see, for instance, [32]) as inputs. This section presents the development from the GNSS equations to the final formula that estimates the combined satellite and receiver DCBs in terms of the GF combinations of the code and the calibrated carrier-phase.

The GNSS observables consist of the linear combination of the geometric distance (between satellite and receiver) and additional propagation delay sources. They can be classified into two groups: dispersive and non-dispersive terms. Namely, the link

between observables, geometric distance and additional delay terms is summarized in the following expression:

$$
\begin{aligned}
P_{r,i}^s &= \rho_r^s + c(d\tau_r - d\tau^s) + T_r^s + I_{r,i}^s + D_{r,i} + D_i^s + \epsilon_i \\
L_{r,i}^s &= \rho_r^s + c(d\tau_r - d\tau^s) + T_r^s - I_{r,i}^s + \frac{c}{f_i}\phi_r^s + B_{r,i}^s + \xi_i ,
\end{aligned}
\tag{2}
$$

where $P_{r,i}^s$ and $L_{r,i}^s$ are the code and carrier-phase (obtained from integration of the Doppler shift) measurements for the $i^{th}$ frequency; $r$ and $s$ stand for the receiver and satellite indexes; $\rho_r^s$ is the geometric distance; $d\tau_r$ and $d\tau^s$ are the receiver and satellite clock errors, respectively; $I_{r,i}^s = \frac{40.3}{f_i^2} \cdot S$, $T_r^s$ and $\phi_r^s$ are the ionospheric delay, tropospheric delay and phase wind-up, respectively, between receiver and satellite transmitter. Note that only the code hardware delays are explicitly indicated for the receiver and the satellite, $D_{r,i}$ and $D_i^s$, respectively, whereas the ambiguity term $B_{r,i}^s$ only affects carrier-phase measurements and it also contains the corresponding carrier-phase hardware delays.

The combination of the code from two different frequencies, for example, corresponding to $L_1$ and $L_2$ carriers, removes all non-dispersive terms, thus yielding the GF combination:

$$
P_{r,GF}^s \equiv P_{r,2}^s - P_{r,1}^s = I_{r,GF}^s + D_{r,GF} + D_{GF}^s ,
\tag{3}
$$

where the associated ionospheric delay $I_r^s \equiv I_{r,GF}^s = \alpha \cdot \int_r^s N_e\, dl$ is proportional to the path integral of the ionospheric electron density, $N_e$, from the receiver $r$ to the satellite $s$; and $D_{r,GF}$, $D_{GF}^s$ are the GF combinations of, respectively, the receiver and satellite hardware delays, a.k.a. differential code biases (DCBs).

Similarly to the GF combination of the code in Equation (3), the GF combination of the carrier-phase is as follows:

$$
L_{r,GF}^s \equiv L_{r,1}^s - L_{r,2}^s = I_{r,GF}^s + c\left(\frac{1}{f_1} - \frac{1}{f_2}\right)\phi_r^s + B_{r,GF}^s ,
\tag{4}
$$

where $B_{r,GF}^s \equiv B_{1,r}^s - B_{2,r}^s$ is the GF combination of the ambiguities in both frequencies and $\phi$ is the phase wind-up angle, with a term impact that is small (a fraction of the difference of wavelengths of both frequencies $\lambda_1 - \lambda_2$) and can be corrected. The replacement of the ionospheric term in Equation (3) using Equation (4) yields, neglecting the code thermal noise and multipath, the following expression:

$$
P_{r,GF}^s = L_{r,GF}^s - B_{r,GF}^s - (\lambda_1 - \lambda_2)\phi_r^s + D_{r,GF} + D_{GF}^s ,
\tag{5}
$$

and, thereby, after rearranging terms, it leads to the following expression of DCBs in terms of GF combinations of the observables and the ambiguity:

$$
D_{r,GF} + D_{GF}^s = P_{r,GF}^s - \tilde{L}_{r,GF}^s ,
\tag{6}
$$

where $\tilde{L}_{r,GF}^s \equiv L_{r,GF}^s - \hat{B}_{r,GF}^s - (\lambda_1 - \lambda_2)\phi_r^s$ is the calibrated carrier-phase of the GF combination. It is computed from the measured carrier phase, $L_{r,GF}^s$, minus the estimated value of the corresponding ambiguity, $\hat{B}_{r,GF}^s$ and the small wind-up term as well, $(\lambda_1 - \lambda_2)\phi_r^s$. In other words, it is the estimated GF ionospheric delay, $I_{r,GF}^s$. Consequently, according to Equation (6), the sum of the transmitter and receiver DCBs can be estimated as residuals from the difference between the GF combinations of the calibrated carrier-phase and the code.

However, in order to further proceed and estimate those DCBs, it is necessary to previously estimate the ambiguity term, $B_{r,GF}^s$. It is possible to do so with further modelling of the ionospheric term in Equation (4). A system of equations based on Equation (4) is rank-deficient. However, it is possible to fix the rank by any model that separates the relative geometry between satellite and receiver and the state of the ionosphere [41]. The reason

is that the ionospheric term, $I_r^s$, as opposed to the ambiguity term, $B_{r,GF}^s$, depends on the relative geometry.

For example, the simplest method would be the so-called thin-shell model, whereby the ionospheric delay is computed as follows:

$$I_{r,GF}^s = \alpha S_r^s = \alpha \left[ 1 - \left( \frac{R_e \cos E_r^s}{R_e + h} \right)^2 \right]^{-1/2} V_{IPP_r^s} \, , \qquad (7)$$

where $S$ is the Slant Total Electron content (STEC) along the path from the transmitter to the receiver, $R_e$ is the average Earth's radius, $h$ is the effective height (e.g., 450 km adopted in IGS, see [33]), where all the ionospheric electron content is assumed located, $E_r^s$ is the elevation angle of the satellite $s$ above the local horizon of receiver $r$ and $V_{IPP_r^s}$ is the Vertical Total Electron Content at the ionospheric pierce point (a.k.a. IPP) where the straight line between the satellite $s$ and the receiver $r$ pierces the thin shell. This model consists of mapping the VTEC along the line of sight, thus computing the corresponding STEC. Note that, as has been discussed above, $STEC$ ($S$) is also the path integral of the ionospheric electron, $N_e$, content along the line of sight between the receiver $r$ and the GNSS satellite $s$, i.e., $S = \int_r^s N_e \, dl$.

Although the simplest, the thin-shell model is also the less accurate one. See, for example, ref. [42] for further improvement of $VTEC$-based models.

Another approach is the estimation of the carrier phase ambiguity $B_{r,GF}^s$; in addition, the ionospheric electron density, $N_e$, follows the approach summarized in Section 2.

Regardless of whether the model is based on $VTEC$ or electron density estimation, both strategies can estimate the ambiguity term, $B_{r,GF}^s$, thus calibrating the GF carrier-phase, $L_{r,GF}^s$, and, eventually, can lead to the computation of DCBs according to Equation (6).

Altogether, TOMION combines and solves the following system of equations:

$$
\begin{aligned}
L_{r,GF}^s(t_k) - (\lambda_1 - \lambda_2)\phi_r^s &= \alpha \cdot \sum_i N_{e,i}(t_k) \cdot l_i + B_{r,GF}^s(t_k) \\
N_{e,i}(t_k) &= \hat{N}_{e,i}(t_{k-1}) + \delta(t_k) \\
B_{r,GF}^s(t_k) &= \hat{B}_{r,GF}^s(t_{k-1}),
\end{aligned}
\qquad (8)
$$

where $t_k$ stand for the $k^{th}$ time-epoch; $\phi_r^s$ is the wind-up angle; $N_{e,i}$ and $\hat{N}_{e,i}$ are the electron density and its estimate, respectively, at the $i^{th}$ voxel; $\delta$ is the system prediction noise; and $B_{r,GF}^s$ and $\hat{B}_{r,GF}^s$ are the ambiguity term and its estimate, respectively. In cases when a cycle-slip occurs, i.e., when a significant jump in $L_{r,GF}^s$ drift rate occurs (a variant of method described in [43]), the Kalman filter starts computing the new ambiguity, after considering the unknown as a white noise random process right during the first epoch after the cycle-slip.

Note that, as a consequence of estimating $B_{r,GF}^s$ on an epoch-wise basis with Equation (8), DCB estimates and their uncertainty can then be estimated epoch-wise as well either in real-time by solving Equation (6), or in post-process by considering the last estimation of the carrier phase ambiguity given by the Kalman filter for each given transmitter–receiver phase continuous arc, the one typically better estimated. This is the way in which UPC-IonSAT provides the DCBs to IGS.

## 4. Data and Results

The dataset consists of carrier-phase (L1W and L2W) and code (C1W and C2W) 1 Hz measurements from the POD GPS receivers on-board the three MetOp LEO satellites A, B and C, for day 176 and days 224 to 252, 2020, and the corresponding receiver and transmitter precise positions.

A first, qualitative way of comparing the performance of the TOMION model Equation (8) is looking at the consistency of the electron density solutions, considering the different combinations between the sources of GPS data (ground-based and MetOp POD) and different vertical resolutions: two vs four layers, centered on the MetOp orbital height (i.e., one and two layers of them, respectively, right above MetOp satellites).

A second, quantitative way of comparing the performance of TOMION, focusing on four vs two layers, is looking at the impact on the estimated standard deviations of the corresponding receiver and transmitter C2W-C1W DCB estimate, $D_{r,GF} + D^s_{GF}$. They are obtained, one per Line-Of-Sight (LOS), after TOMION directly estimates the carrier phase ambiguity at the same time as the electron density. DCBs are expected to be rather stable within hours, or maybe even days.

Therefore, if the number of layers (e.g., two or four) is not relevant, then the DCBs and their uncertainties estimated by each model should be rather the same values (i.e., below the SNR of the GPS signal). In this regard, this section presents an analysis of the estimated DCBs that demonstrates that the number of layers has an impact on the accuracy of the ionospheric sounding.

The stability of the DCBs estimated by TOMION, from MetOpt POD GPS measurements only and over four consecutive weeks was analyzed. The DCB estimation was assessed by comparing with GPS transmitter DCBs computed with different techniques by CODE and JPL IGS IAACs (Ionosphere Associate Analysis Centers) from ground-based measurements only.

It has been previously reported that the results computed by TOMION with ground receivers only might be affected by vertical correlations when the ionospheric grid is designed with more than two layers [29]. This is due to the relative geometry between satellites and ground receivers. Indeed, GNSS measurements from a ground receiver provide high horizontal resolution, but they lack vertical resolution. However, the MetOp POD GPS measurements from the MetOp orbit height (800 km approximately) provide certain vertical resolution, up to two layers above, thereby removing the vertical correlation artifacts in four-layer models.

### 4.1. Ionospheric Tomography

We compared, similar to what we did in [38], the tomographic solution under different combinations of sources of dual-frequency GPS measurements, mainly 150 ground GPS receivers, providing ambiguous STEC observations from the GPS transmitters (around 20,200 km height) to the ground and GPS measurements gathered from the POD receiver on board MetOp A, B and C.

A representative example can be seen in Figure 1, where the Vertical Upper Electron Content above 780 km (VUEC) provides more consistent results when the MetOp measurements are considered. In the second and third rows, the negative VUEC values at very high latitudes are negligible, compared with the runs with only ground-based GPS measurements, the first row (it has to be considered that no background model or any constraint on electron density positivityis considered). This improvement is still more clear when two layers above MetOp satellites (four layers in total, second column) are considered instead of just one (first column). The tomographic results with MetOp POD GPS-only (central-right plot at Figure 1) are similar in consistency to the ones with ground and MetOp POD GPS data, just showing a reduced range of values due to the smaller set of illuminated voxels (bottom-right plot at the same figure).

### 4.2. DCBs of MetOp POD GPS Receivers

In Figure 2, the TOMION daily estimations of the GPS $P_2 - P_1$ DCBs of the POD GPS receivers of the three MetOp LEO satellites referring to the average of the GPS transmitter DCBs (taken as data as usual in IGS) are plotted vs time.

On the one hand the TOMION estimation shows stability within the 28-day period, peak-to-peak, better than 1 TECU, i.e., 0.3 ns. This stability is similar or better than the typical stability around 1 ns observed for ground GPS receivers (e.g., see Figure 16 in [33]).

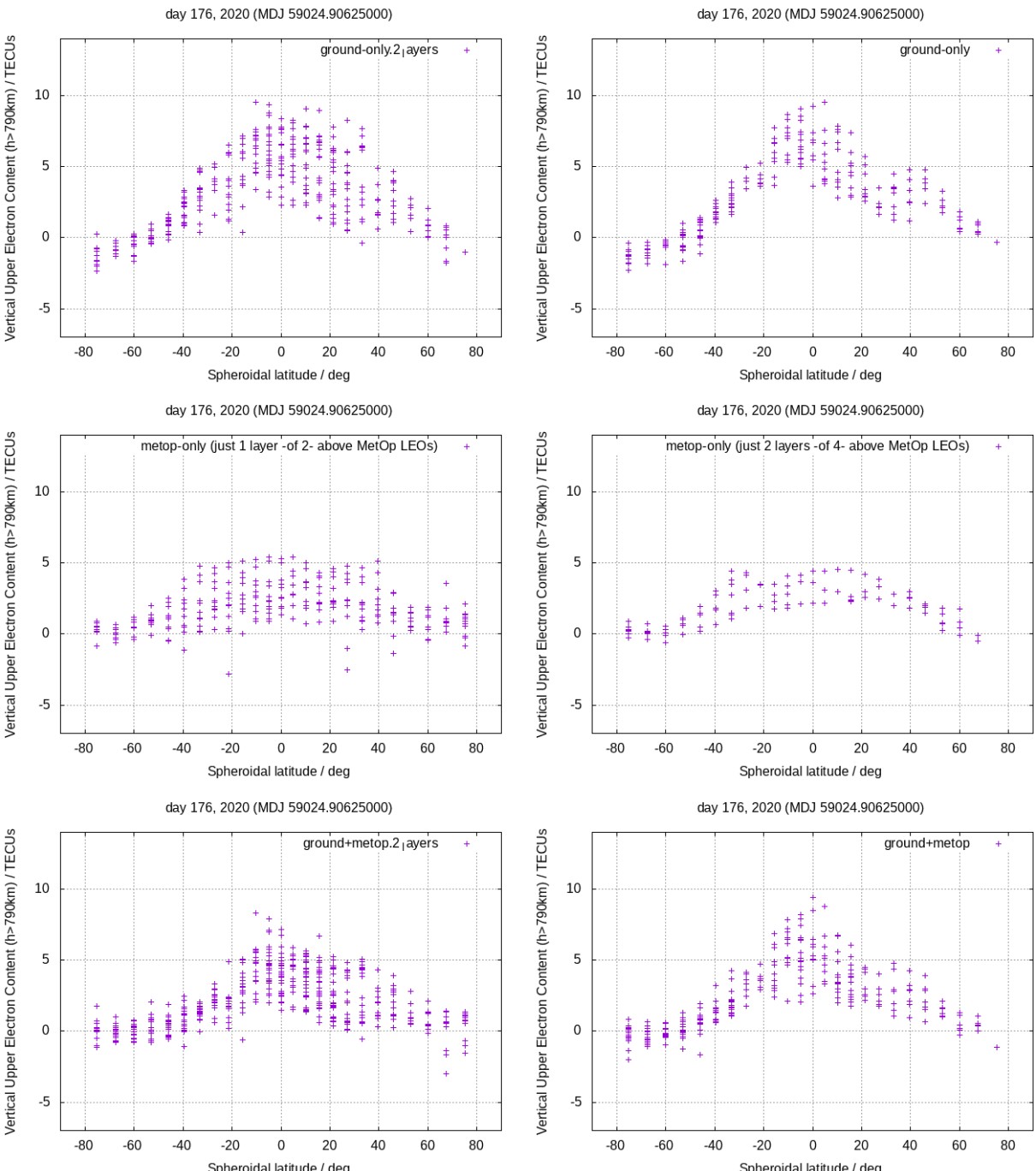

**Figure 1.** Comparison of the Vertical Upper Electron Content (in TECUs) vs latitude at GPS time 21 h 45 m of day 176 of year 2020, considering: (**left**) 2 layers (1 above MetOp LEO satellites) in the first column and (**right**) 4 layers (2 above MetOp LEO satellites) in the second one. The input measurements consist of ground GPS ground-based measurements only (first row), MetOp POD GPS measurements only (second row) and combined ground GPS and MetOP POD GPS measurements (third row).

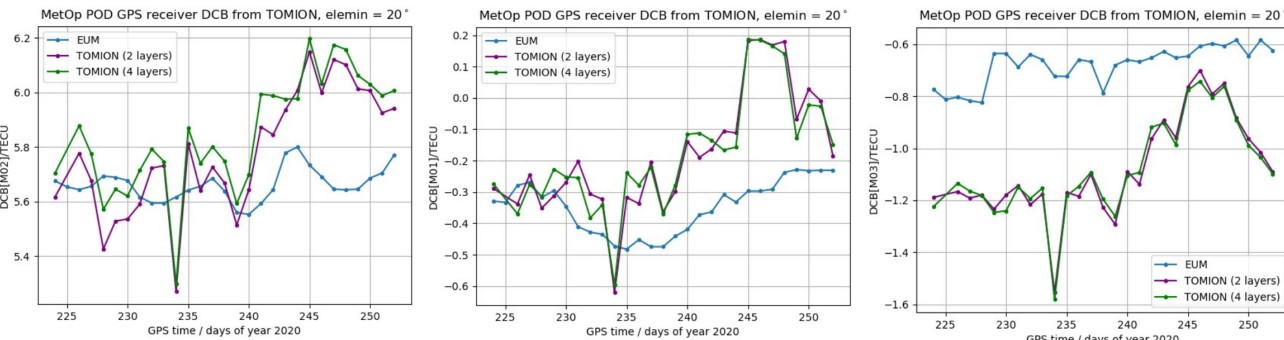

**Figure 2.** Daily values of the MetOp POD GPS receiver DCBs, in TECUs, from left to right, DCB[meta], DCB[metb], DCB[metc], in TECU, vs. day of year 2020, computed with TOMIONv1 with MetOp GPS data only under 2-layer TOMION run (green points), under 4-layer TOMION model (magenta points) and with the updated EUMETSAT technique (blue).

In detail, the stability of TOMION results with four layers (two above the MetOp LEO satellites) are slightly more stable (at below 0.1 ns level, see left plot in Figure 2) than the results with two layers only, namely it is 1 above. Also note that the tomographic DCB results vary in time. Although we cannot identify the origin of such DCB variability, it has been previously reported that GNSS DCB estimates vary with environmental [11] and/or CPU temperature [44]. Furthermore, if DCB and temperature are correlated as presented in [11] for ground receivers, then it might be possible to model the environmental temperature variability with DCB estimates at LEO orbit heights.

In Figure 2 the new EUMETSAT DCB results, after applying the zero TEC technique as discussed by [44], are also shown, comparing well with the TOMION results. The EU-METSAT approach considers as daily receiver DCB the first quartile of the distribution of the minimum apparent sTEC when they are computed with zenith data over the previous 5 days: only data at high latitudes (above 60 deg north/south), during night local times (from 18:00 to 6:00) and taken above 40 deg elevation, are considered. We can see from Figure 2 that EUMETSAT results seem more stable. This is simply related to the fact that, for estimating the daily DCB for day X, we are taking the first quartile of the minimum sTEC computed for the interval [day X - 4 days; day X]. So, this clearly impacts the stability of the EUMETSAT DCB time series, which are somehow more "smoothed". However, all of them show a variability within 0.3 TECU, without apparent differences between the three Metops.

### 4.3. External Assessment via Transmitter DCBs

It is also possible to assess the TOMION performance (two and four layers) from MetOp GPS POD data only and without any background model, with a comparison between the corresponding GPS transmitter DCB estimates and those from IGS centers, computed from GPS ground data only and regarding the data given by the sum of transmitter DCBs (daily satellite DCBs can be obtained from Equation (6), see Appendix A for further details).

As can be seen in Figure 3, the GPS transmitter DCBs estimated by TOMION, as a by-product of the tomography run from the MetOp A, B and C dual-frequency carrier phase measurements only, are in good agreement with those from CODE and JPL across days 224 to 252, 2020. They typically agree at a few tenths of ns level and up to 0.5 ns in the worse case, i.e., at the level of the agreement shown between transmitter DCBs provided by different IGS ionospheric analysis centers from ground GPS data only (see, for instance, Figure 7 in [33] and associated comments).

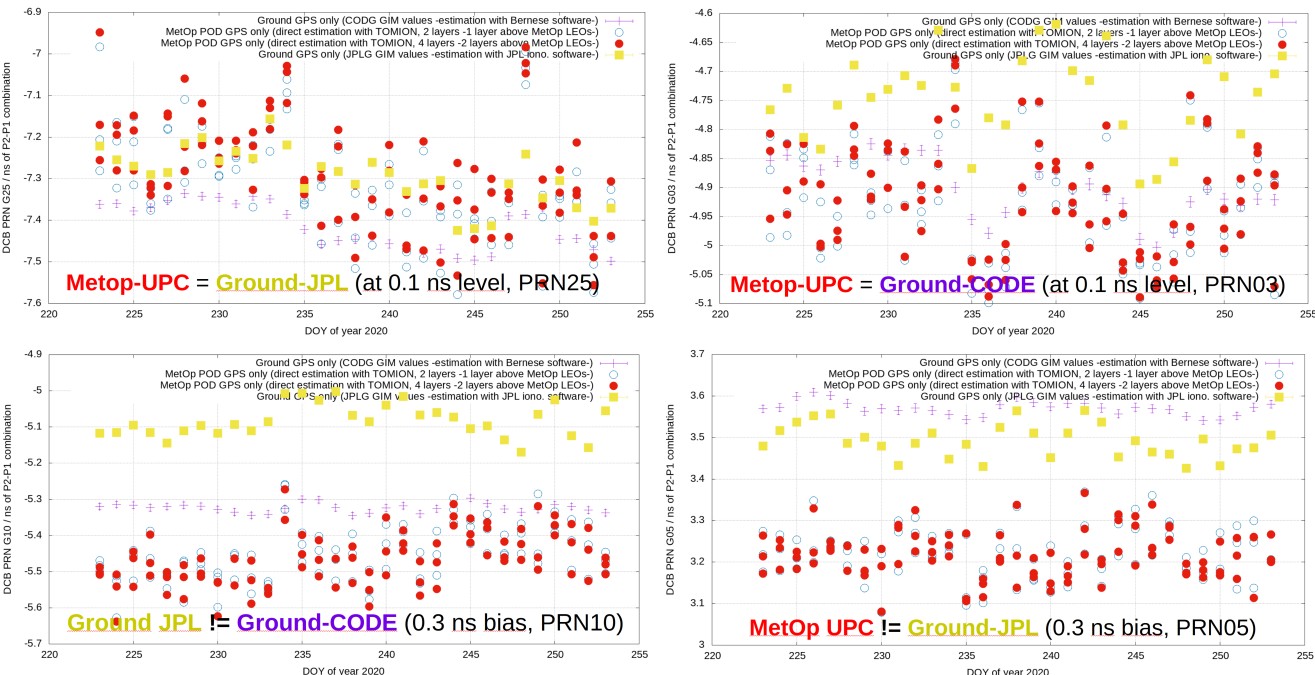

**Figure 3.** Four typical examples of comparison of GPS transmitter DCBs obtained by TOMION runs based on MetOp LEO data only (2 and 4 layers, i.e., 1 and 2 layers above) vs. ground-based values from IGS centers (CODE and JPL): PRN25 (**top−left**), PRN03 (**top−right**), PRN10 (**bottom−left**) and PRN05 (**bottom−right**).

## 5. Discussion

In the previous section, we showed that the estimated tomographic model of the LEO topside electron content only fed with raw GPS ionospheric $L_1 - L_2$ measurements and solved without any background model is able to provide directly consistent electron content results. Moreover, as a byproduct, it is able to provide transmitter DCBs comparable to the ones computed from hundreds of GPS ground-based receivers by other analysis centers.

In this section, we focus on the influence of two particular aspects of the modelling which are not usually discussed when topside electron content is estimated and the DCBs of LEO-based GNSS receivers are compared: the influence of changes in the DCB datum and the GPS receiving antenna phase center variation. Moreover the potential influence of the difference between the LEO orbits will also be addressed.

### 5.1. Influence of Changes in the DCB Data

The estimated DCBs (for GPS in particular) delivered by many ionospheric analysis centers, such as those of IGS (refs. [31,33]) refer to the average of the GPS transmitter values, which is defined as zero. This is the data in the definition of GPS DCBs. As can be seen in the top plots of Figure 4 corresponding to the four analyzed weeks, it might happen that some days some analysis centers (JPL and UPC in this) do not provide the daily DCB values for the complete set of GPS transmitters (top-right plot), but keep the DCB data to the given set of GPS transmitters (top-left plot). However, the effect is small, at or below 0.1 ns at C2W-C1W level, as can be seen, for instance, for GPS10 DCB estimation via JPL in the bottom-left plot of Figure 3 during days 233–243 and 249–250 of 2020, where the JPL DCB datum is based on 30 and not 31 GPS transmitters (top-right plot of Figure 4).

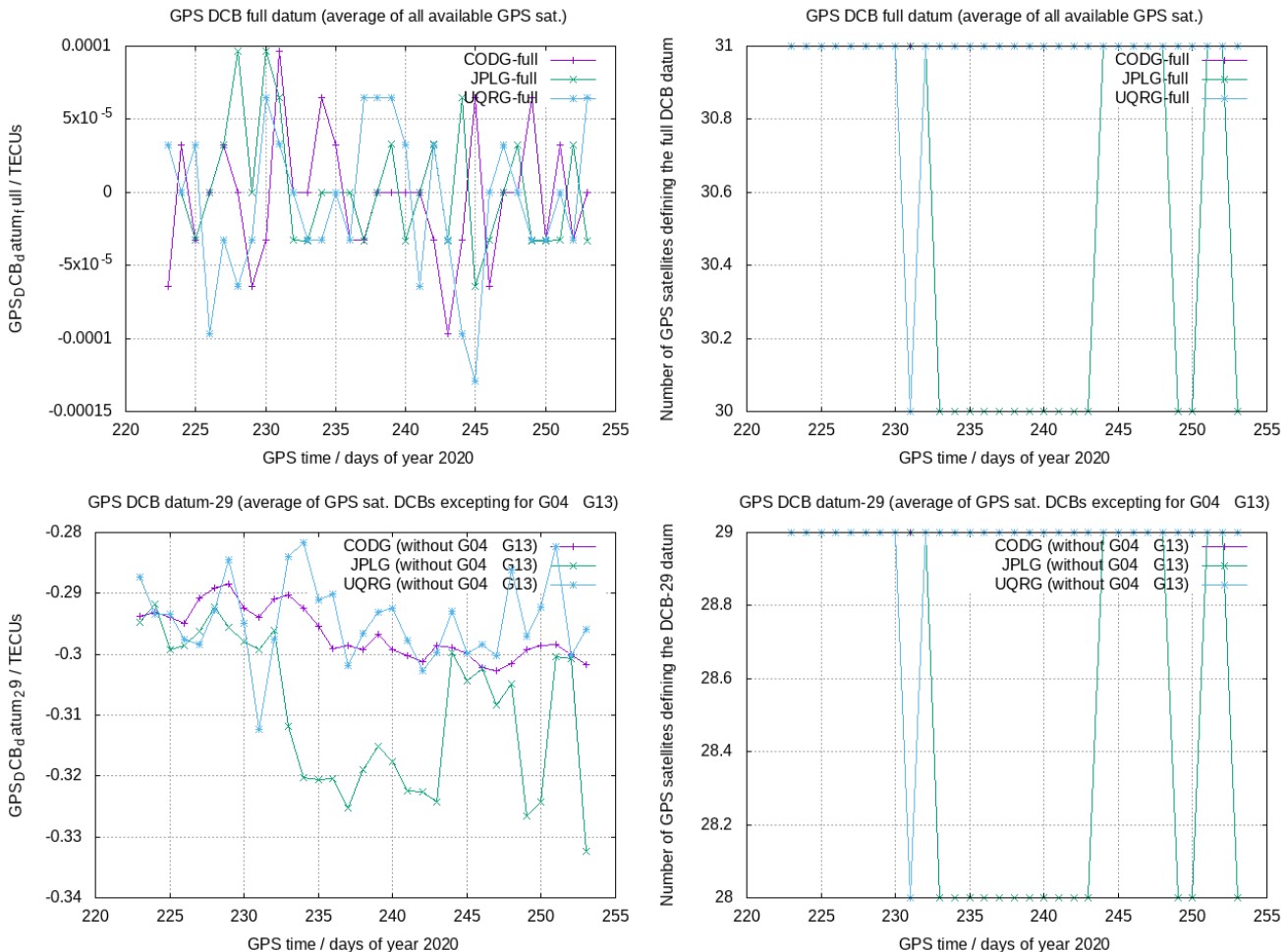

**Figure 4.** Average of all the available GPS transmitter C2W-C1W DCBs (i.e., GPS data) in the CODE, JPL and UPC-IonSAT GIMs (**top−left**); corresponding number of GPS transmitters (**top−right**). Similarly, for a slightly different datum of 29 transmitters (DCB average excluding GPS04 and GPS13), the GPS DCB data (**bottom−left**) and the corresponding number of GPS transmitters among the 29 ones (**bottom−right**).

Similarly, in the estimation of the DCBs based only on MetOp measurements presented in previous sections, two GPS transmitters (G04 and G13) were not available with enough high elevation, from the close orbits of the three LEOs, meta, metb and metb (see Figure 8), and they did not take part in the DCB data of our results, based on the remaining 29 GPS transmitters. Similarly as in the previously noted case of JPL DCBs, the impact is small, at the level of 0.1 ns, i.e., 0.3 TECU approximately. This can be seen in the average of these 29 GPS transmitter DCBs from CODE, JPL and UPC-IonSAT ground-based DCBs, associated with the corresponding GIMs, in the top-bottom plot of Figure 4, where the change of the data for 28 transmitters mostly for JPL (days 233–243 and 249–250) and UPC-IonSAT (day 231) appears in this case as a sort of second-order datum effect. A similar case of loss of one additional GPS transmitter, but during day 234, affects the DCB data in the MetOp-only based solution and explains the jump of -0.3 TECU of the receiver DCB value during this day (see Figure 2).

### 5.2. Influence of the GPS Receiving Antenna Phase Center Variation

Accurate knowledge of the LEO receiver antenna phase center variation can require dedicated in-flight research, such as the case of the characterization of the zenith GPS antenna of MetOp receivers, meta, metb and metb [45]. Despite the horizontal separation of $L_1$ and $L_2$ phase centers being larger than 10 cm, the vertical one is smaller than 1 cm.

This brings us to neglect this correction, assuming that its impact should be very small, because the results in our problem should depend mostly on the close to vertical observations from the LEOs. Indeed, the main correction affects the low elevation measurements (see Figure 5) and the final impact is very low in the tomographic solution (at sub-TECU level, see Figure 6) and in the DCB estimation, typically less or much less than 0.1 ns in C2W-C1W DCB (see representative example for GPS10 in Figure 7).

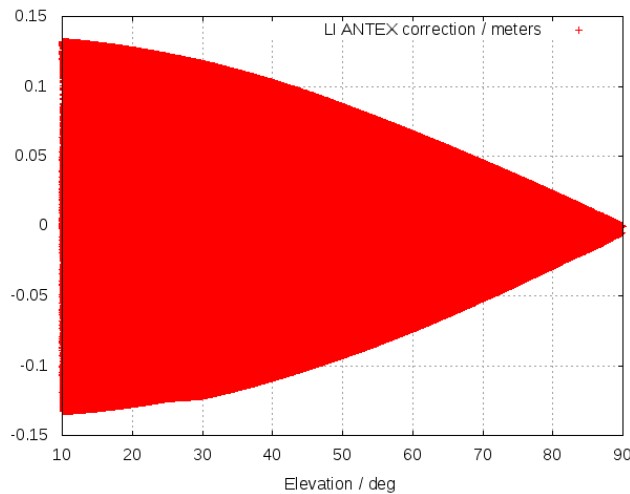

**Figure 5.** GPS receiving antenna corrections for the MetOp LEOs in ionospheric combination during day 176 of year 2020.

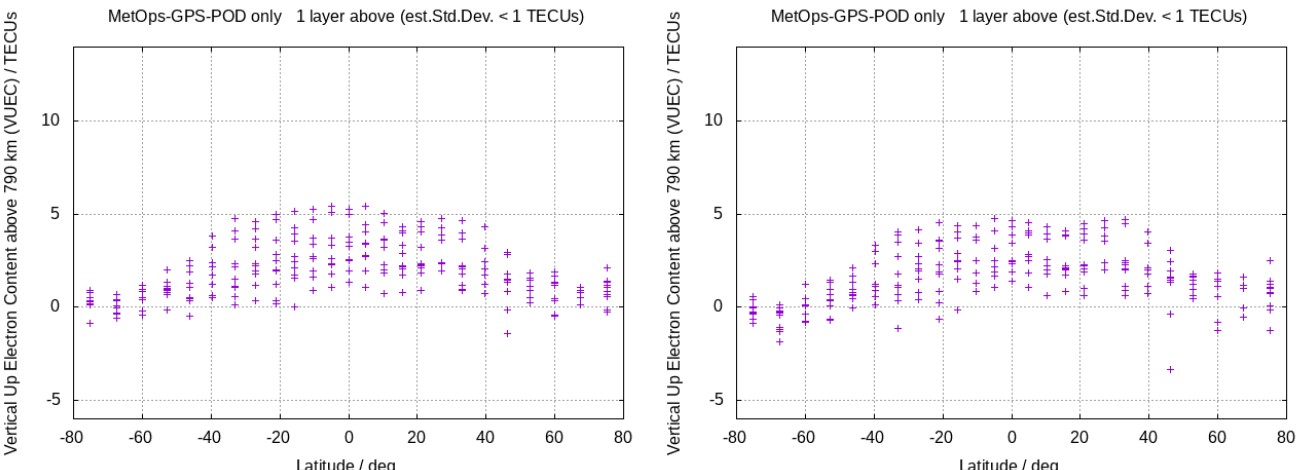

**Figure 6.** Comparison of the Vertical Upper Electron Content (in TECUs) vs latitude at GPS time 21 h 45 m of day 176 of year 2020, considering 2 layers (1 above MetOp LEO satellites) and MetOp POD GPS measurements only: (**left**) Without receiving antenna corrections and (**right**) with receiving antenna corrections.

### 5.3. Potential Influence of the Difference between the LEO Orbits

The distribution of the orbits of the three MetOp LEOs, flying at a similar height around 800 km, are identical for metb and metc and similar for meta, but placed at distances of thousands of km between the LEOs (see Figure 8). However, looking at the different estimation of the transmitter DCBs from the measurements separately of meta, metb and metc (Figure 7), or directly comparing the receiver DCBs (Figure 2), it can be seen than there is no evident relationship between the LEO satellite relative spatial location and them (in fact it might seem more similar to the time evolution of meta and metc DCBs, which do not follow the same orbit).

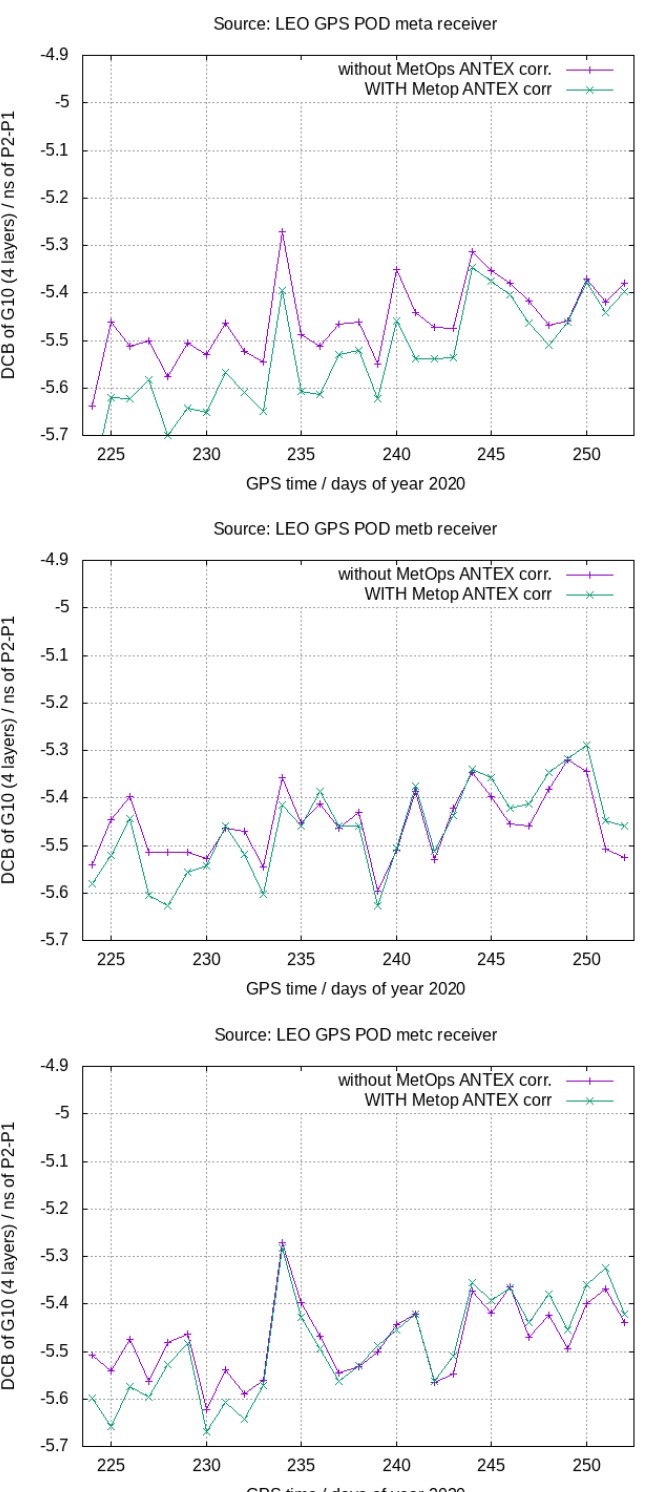

**Figure 7.** Comparison of the daily GPS10 transmitter DCBs estimated from MetOp zenith measurements only without (magenta) and with (green) receiving antenna corrections, two layers above LEOs and since days 224 to 252 of year 2020, from meta (**top**), metb (**center**) and metc (**bottom**) LEOs.

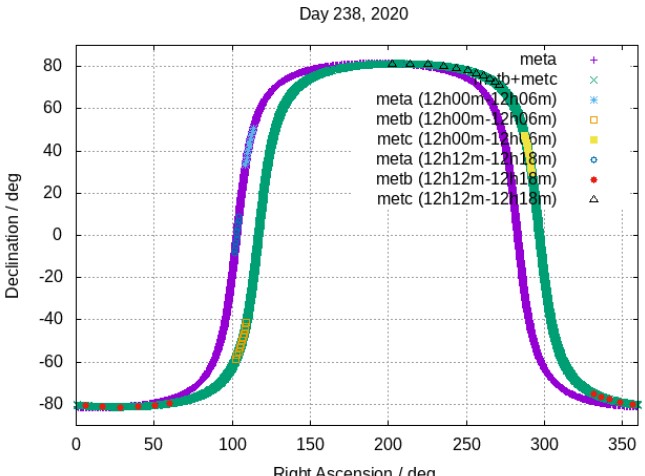

**Figure 8.** Footprint of the orbits of meta, metb, metc MetOp LEOs, for day 238 of year 2020, showing the displacement between 12 h 00 m–12 h 06 m and 12 h 12 m–12 h 18 m.

## 6. Conclusions

The characterization of the GPS carrier-phase-based only tomographic estimation of the LEO topside electron content was summarized. It was demonstrated on a single representative day (176, 2020) for different combinations of input data (among ground GPS and MetOp POD GPS) and different degrees of tomographic resolutions (four layers vs two layers). Slightly better results were obtained with ground GPS + MetOp POD GPS data and with MetOp POD GPS only and showing a stability of the MetOp POD GPS receiver DCBs at a few tenths of the TECU level.

The study was extended to 29 consecutive days (days 224 to 252, 2020) showing a consistent result of the TOMION DCBs, obtained in daily independent runs from carrier-phase data only (stability of the DCB differences among MetOp POD GPS receivers at 0.1 TECU level). The associated transmitter DCBs from MetOp-only data are consistent with the IGS ground-based ones at a few tenths of ns in a P2-P1 delay level, as expected for transmitter DCBs

In this way, a novel method for DCB estimation has been introduced. DCBs are estimated as residuals out of the difference between the GF combinations of the code and the calibrated carrier-phase. The carrier-phase has been calibrated by a tomographic model, TOMION, that decorrelates the ionospheric delay and the ambiguity term. The DCBs and their uncertainty are estimated on an epoch basis, thus showing the variability in time of DCBs, which is consistent with previous works from other authors. The method has been assessed with external GPS transmitter DCBs computed by CODE and JPL and their relative differences are within 0.5 ns. The small influence of some modelling aspects which are not frequently considered, such as the change of DCB data and the influence of the LEO receiver zenith antenna phase variations, were also discussed in detail.

Finally, potential future follow-on research could analyze the existence of a correlation between LEO DCBs and temperature. If such a correlation exists for LEO satellites, then it would provide the possibility of using DCBs as a proxy of atmospheric temperature variability.

In conclusion, high accuracy estimation of LEO receiver DCBs provides a method for monitoring the state of GNSS receivers and for a quick and accurate computation of the STEC corresponding to the POD GPS receiver lines-of-sight. This approach has as a main advantage providing a simultaneous estimation of the LEO topside ionospheric electron content distribution and of the transmitter and receiver DCBs—everything in an autonomous way (i.e., only based on raw GPS zenith LEO observations). This is done in a simple manner, where the DCBs are computed as an a posteriori product of the ionospheric tomography (then suitable for multi-GNSS measurements) and without any background model or initial set of DCB values required.

**Author Contributions:** Conceptualization, M.H.-P. and R.N.; methodology, M.H.-P.; software, M.H.-P. and V.G.; validation, M.H.-P., R.N. and L.Y.; formal analysis, M.H.-P.; investigation, M.H.-P.; resources, R.N.; data curation, R.N., G.O.-P. and M.H.-P.; writing—original draft preparation, G.O.-P. and M.H.-P.; writing—review and editing, M.M.H., F.S.P., L.Y., R.N., V.G., G.O.-P. and M.H.-P.; visualization, M.H.-P.; supervision, R.N., M.M.H., F.S.P. and M.H.-P.; project administration, R.N., M.M.H. and M.H.-P.; funding acquisition, M.M.H. and M.H.-P. All authors have read and agreed to the published version of the manuscript.

**Funding:** This research was funded by EUMETSAT in the context of the project *Assessment of GRAS Ionospheric Measurements for Ionospheric Model Assimilation* (GIMA, number EUM/CO/21/4600002530/RN).

**Data Availability Statement:** The external DCB data are openly accessible (https://cddis.nasa.gov/archive/gnss/products/ionex accessed on 5 November 2021) from Crustal Dynamics Data Information System. The input GPS measurements of the MetOp POD GPS receivers and precise MetOp orbits, can be solicited to EUMETSAT if needed.

**Acknowledgments:** The work has been performed coinciding with the execution of the PITHIA-NRF H2020 project (H2020-INFRAIA-2018-2020101007599).

**Conflicts of Interest:** The authors declare no conflict of interest.

**Appendix A**

In order to obtain the transmitter DCB, firstly we average Equation (6) over time, thus yielding the following expression for the daily total DCBs, $\bar{D}_r^s$:

$$\overline{D_r^s} \equiv \overline{D_{r,GF} + D_{GF}^s} = \frac{1}{N} \sum_{i=1}^{i=N} P_{r,GF}^s(t_i) - \tilde{L}_{r,GF}^s(t_i), \tag{A1}$$

where $N$ is the number of epochs and $t_i$ the time at the $i^{th}$ epoch. Secondly, the daily receiver is computed by averaging $\overline{D_r^s}$ over satellites:

$$\hat{D}_r = \frac{1}{M} \sum_{i=1}^{i=M} \overline{D_r^i}, \tag{A2}$$

where $M$ is the number of satellites in view by the receiver. Now, notice that $\overline{D_r^i} = \overline{D_r + D^i} = \overline{D_r} + \overline{D^i}$, then:

$$\hat{D}_r = \frac{1}{M} \sum_{i=1}^{i=M} \left[ \overline{D_r} + \overline{D^i} \right], \tag{A3}$$

which, after setting the sum of daily satellites DCBs to zero, i.e.,

$$\sum_{i=1}^{i=M} \overline{D^i} = 0, \tag{A4}$$

leads to:

$$\hat{D}_r = \frac{1}{M} \sum_{i=1}^{i=M} \overline{D_r}, \tag{A5}$$

i.e., the daily satellite-averaged receiver DCBs. Finally, subtraction of $\hat{D}_r$ in Equation (A1) yields the daily satellite DCB, $\hat{D}^s$, as follows:

$$\hat{D}^s = \overline{D_r^s} - \hat{D}_r. \tag{A6}$$

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
