# Peer review of "Topside Ionospheric Tomography Exclusively Based on LEO POD GPS Carrier Phases: Application to Autonomous LEO DCB Estimation"

_remotesensing, doi:10.3390/rs15020390_

Round 1

Reviewer 1 Report

It is of great value to study the method of estimating DCBs from GPS transmitters and receivers on-board LEO satellites. The authors applied TOMION to DCB estimation and did some relevant analysis. The performances of the above method are assessed, including the stability of the TEC solution and the accuracy of the DCB product. From the experimental results, the number of layers used for ionospheric tomography has some influence on the ionospheric sounding. And the addition of LEO satellite observations reduces the number of TEC zero-values at high latitudes. Last but not least, the estimation of satellite DCB based on LEO satellite data is in good agreement with IGS ionospheric products. I hope the following comments can be helpful in improving the paper.

(1) I have noted that the work was not submitted with reference to the authors' instructions,as mentioned in ‘https://www.mdpi.com/journal/remotesensing/instructions’, which required an article length of at least 18 pages for submissions to RS (the paper is only 11 pages). This paper is too short and suggests showing more experimental results and statistical information.

(2) To be more specific, the authors should indicate the type of signal used for DCB estimation, e.g., C1C/C2W.

(3) Since the fast movement of the receiver leads to more likely cycle slips and outliers in the GPS observations of LEO. The authors have given the elevation cutoff in Figure 2, please also point out the cycle slip detection method.

(4) The font in all figures is too small making it difficult to read.

(5) The unavailability of some satellites on some days causes a change in the central datum of the DCB estimate, which causes a systematic variation in the DCB values for individual days from others. Is there such a situation? How to deal with it, if any?

(6) For the receiver DCBs, I notice that MetOp B and MetOp C are closer, while MetOp A differs from both, is it because the space environment or hardware of MetOp A is different from both? Please give a brief analysis.

Author Response

===> Answers to points of Reviewer #1:

>>It is of great value to study the method of estimating DCBs from GPS transmitters and receivers on-board LEO satellites. The authors applied TOMION to DCB estimation and did some relevant analysis. The performances of the above method are assessed, including the stability of the TEC solution and the accuracy of the DCB product. From the experimental results, the number of layers used for ionospheric tomography has some influence on the ionospheric sounding. And the addition of LEO satellite observations reduces the number of TEC zero-values at high latitudes. Last but not least, the estimation of satellite DCB based on LEO satellite data is in good agreement with IGS ionospheric products. I hope the following comments can be helpful in improving the paper.
>>

Many thanks for the comments and inputs. Please be so kind to consider the changes tracking version of the updated manuscript to find the modifications commented below.

>> 
>>
>>(1) I have noted that the work was not submitted with reference to the authors' instructions,as mentioned in ‘https://www.mdpi.com/journal/remotesensing/instructions’, which required an article length of at least 18 pages for submissions to RS (the paper is only 11 pages). This paper is too short and suggests showing more experimental results and statistical information.
>>
>> 

Please find the new section "Discussion", included following as well the editor's suggestion, providing additional details.

>>
>>(2) To be more specific, the authors should indicate the type of signal used for DCB estimation, e.g., C1C/C2W.
>>

Yes, sorry. The GPS signals and the estimation of the C2W-C1W DCB is now explicitely indicated in section "Data and results". Thank you.

>> 
>>
>>(3) Since the fast movement of the receiver leads to more likely cycle slips and outliers in the GPS observations of LEO. The authors have given the elevation cutoff in Figure 2, please also point out the cycle slip detection method.
>>

Now the cycle-slip detection method used, the Blewitt's method, is referenced in the last part of section 3, indicating the corresponding reference.

>> 
>>
>>(4) The font in all figures is too small making it difficult to read.
>>

Now the figures are reproduced at a significantly larger scale. Thank you for letting us know it.

>> 
>>
>>(5) The unavailability of some satellites on some days causes a change in the central datum of the DCB estimate, which causes a systematic variation in the DCB values for individual days from others. Is there such a situation? How to deal with it, if any?
>>
>> 

The small impact when a difference in the DCB datum occurs is now commented in the new section "Discussion", giving two examples from our estimation with TOMION and MetOp GPS POD measurements only, and from JPL GIMs.

>>
>>(6) For the receiver DCBs, I notice that MetOp B and MetOp C are closer, while MetOp A differs from both, is it because the space environment or hardware of MetOp A is different from both? Please give a brief analysis.

We are a bit confused about this question.

If the question relates the absolute values of the DCB time series, the values are different because of the slightly different hardwares. And probably the HW of the Metop-B/C receivers are more similar to the one of Metop-A (which might also suffer of aging, since it was orbiting for almost 17 years).

Another thing is that Metop-A was at the end of his life and started to deorbiting. So its orbit was already drifting from the Metop-B/C ones. And this is shown in the new Figure 8. But, as explained in the new section 5.3, the impact of the different Metop-A orbit should be negligible.

But, just to clarify, we really don’t see such a big different behaviours between the three DCBs time series. Regarding the TOMION results, a part for the jump for day 234 impacting all the Metops DCBs, all of them show quite “constant” DCBs until Day 240, and then all of them show an increase followed by a decrease in the remaining time interval.

What we can see from the Figure 2 is that EUM results seems more stable. This is simply related to the fact that, for estimating the daily DCB for day X, we’re taking the 1st quartile of the minimum sTECs computed for the interval [day X – 4 days; day X]. So, this clearly impacts the stability of the EUM DCB time series, which are somehow more “smoothed”. But all of them show a variability within 0.3 TECU, without apparent differences between the three Metops.

Please find our inputs to this point at the end of section 4.2 and in the new subsection 5.3, within "Discussion".

Author Response

===> Answers to points of Reviewer #2 (indicated in file 'peer-review-24214436.v3.pdf'):

>>A few studies have already developed tomography approaches to obtain topside electron density representations, but a common approach is to calibrate the DCB unknowns before the tomography inversion.
>>
>>Actually, this work in this area has previously been studied by some scholars, such as Fuproposed an approach to restruct the 3D electron density of ionosphere which the DCB and the electron was estimated together.

Thank you for your inputs, in particular for mentioning Hu et al. (2019), which is another relevant reference; now it is cited in the penultimate paragraph of section 1. Please be so kind to consider the changes tracking version of the updated manuscript to find the modifications commented below.

>>
>>The paper introduce the Kalman filter to reconstruct the ionospheric structure, and the electron density is estimated as random walk. How do you determine the background covariance matrix ,the process noise matrix and observation covariances ?

The random process is estimated with an update of the covariance matrix which approximates the one of the random walk, with a process noise of 9.E-8 ( meters of L1-L2 delay / km )^2 / hour, everything optimized empirically, after 25 years of daily processing for contributing to IGS with the UPC-IonSAT GIMs, including different experiments. The initial observation covariances is assumed diagonal, with an apriori standard deviation per L1-L2 observation equation of 0.05 meters.

This information has been incorporated in the penultimate paragraph of section 2.

>>
>>what is the available STEC that provided by radio occulation? Is it calibrated TEC or podTec?
>>

No calibrated STEC is used as input for solving the problem, just the raw GPS measurements of the zenith GPS antenna of the 3 MetOp LEOs, as it is now indicated in the penultimate paragraph of section 1.

>>[1] Fu N, Guo P, Wu M, et al. The Two-Parts Step-by-Step Ionospheric Assimilation Based on Ground-Based/Spaceborne Observations and Its Verification[J]. Remote Sensing, 2019, 11(10).

This reference has been incorporated, as it has been indicated above.

Reviewer 3 Report

The article is quite interesting, presents a novel technique to estimate DCBs from GPS transmitters and receivers on-board Low Earth Orbit (LEO) satellites. In my opinion, it can be approved for publication after minor corrections.

Detailet comments:

In many places in the paper, 3 or more references appear in one statement (e.g.: ([Yue et al., 2011], [Zhang and Tang, 2014]; [Lin et al., 2016]); ([Heise et al., 2002]; [Spencer and Mitchell, 2011]; [Wu et al., 2015]; [Kim et al., 2018];[Prol and Hoque, 2022]); ([Hern´andez-Pajares et al., 2009], [Hern´andez-Pajares et al., 2017], [Roma-Dollase et al., 2018]); …) only with a brief note of what they concern. Please describe the cited articles in more detail (few sentences) or remove them if they are not necessary.

Page 2.

Please explain what „a.k.a.” means.

jth voxel” or „j-th voxel”?

The descriptions next to all the figures are too small which makes them unreadable. Please correct this.

Please describe (in Conclusions) more clearly/understandably the advantages and disadvantages of the proposed method in relation to existing ones.

Author Response

===> Answers to points of Reviewer #3:

>>The article is quite interesting, presents a novel technique to estimate DCBs from GPS transmitters and receivers on-board Low Earth Orbit (LEO) satellites. In my opinion, it can be approved for publication after minor corrections.
>>

Thank you for your comments. Please be so kind to consider the changes tracking version of the updated manuscript to find the modifications commented below.

>> 
>>
>>Detailet comments:
>>
>>In many places in the paper, 3 or more references appear in one statement (e.g.: ([Yue et al., 2011], [Zhang and Tang, 2014]; [Lin et al., 2016]); ([Heise et al., 2002]; [Spencer and Mitchell, 2011]; [Wu et al., 2015]; [Kim et al., 2018];[Prol and Hoque, 2022]); ([Hern´andez-Pajares et al., 2009], [Hern´andez-Pajares et al., 2017], [Roma-Dollase et al., 2018]); …) only with a brief note of what they concern. Please describe the cited articles in more detail (few sentences) or remove them if they are not necessary.
>>

Please find below the next updates, i) to v), following your suggestion:

i) Lines 18-19. The following paragraph

“(...) are encoded. For example, [1] quantified timing delays between the different legacy and modernized GPS and Galileo signals broadcast on L1 and their dependencies on factors like user receiver filter bandwidth, filter transfer function, and delay locked loop (DLL) correlator spacing; [2] provided a comprehensive overview of pseudorange biases and their dependency on receiver front-end bandwidth and correlator design; and [3] showed that the long-term variations of current DCB products may vary significantly,.”
is replacing

“(...) are encoded ([1]; [2], [3])”.

ii) Line 25. The following text

“Over the last two decades, however, significant research concerning the Global Positioning System (GPS) DCB has been conducted ([8]; [9]; [10]), finding intraday variabilities with clear correlations with temperature ([11], [12]) and possibilities to estimate DCB on-board low Earth orbit (LEO) satellites ([13], [14]; [15]).“

has been replaced with the following text:

“Over the last two decades, however, significant research concerning the Global Positioning System (GPS) DCB has been conducted. For example, [8] analyzed the stability of the GPS instrumental biases; [9] found that the day-to-day and annual variation of the estimated GPS DCB is related to the ionospheric variability; and [10] concluded that DCBs variability is attributed to the GPS satellite replacements with different satellite types and the zero-mean condition imposed on all satellite DCBs. Furthermore, [11] and [12] found intraday variabilities with clear correlations in temperature. Estimate methods for DCBs have also been developed for on-board low Earth Orbit (LEO) satellites. In this regard, [13] showed that the STEC estimate might be enhanced if the temperature dependency of DCB estimation is considered as well. [14] estimated the COSMIC GPS DCB receivers in order to perform plasmaspheric observations from LEO satellites with GNSS data. Moreover, [15] developed a method whereby GNSS observations from standalone LEO satellites (i.e. with no ground-based GNSS data) can estimate GPS satellite DCBs.”

iii) Line 45: The text

“A few studies have already developed tomography approaches to obtain topside electron density representations ([22]; [23]; [24]; [25]; [26]), but a common …”

has been replaced with

“A few studies have already developed tomography approaches to obtain topside electron density representations. For example, [22] presented initial results of the sounding of the topside ionosphere and plasmasphere based on GPS measurements from CHAMP. [23] introduced a new mathematical approach to imaging the electron density distribution in the high regions of the topside ionosphere and the plasmasphere using GPS measurements from LEO satellites. A data assimilative method based on 3-D Var for the sounding of the ionosphere and plasmasphere using COSMIC-GPS measurements was introduced in [24]. Using Jason-1 plasmaspheric total electron content (TEC) measurements, [35] developed a tomographic technique for the reconstruction of the plasmaspheric density. COSMIC satellite data has also been used for a tomographic method that estimates the plasmasphere [26]. However, the common…”

iv) Line 83: The text

“...GIM in IGS ([33], [34], [35]).”

has been replaced with the following text:

“...GIM in IGS [33]. UQRG GIM has been assessed with other GIMs and it provides RMS values of 2 TECU [34], [35].”

v) Line 171: The following text

“...provides the DCBs to IGS, see [33], [32], [34] and [35].”

hast been replaced with the following text:

“...provides the DCBs to IGS.”

>>
>>Please explain what „a.k.a.” means.

Sorry; it means "also know as". Now the meaning is explicitely indicated in the first appearance, right before equation (1).

>>
>>„jth voxel” or „j-th voxel”?
>>

Sorry for the typo; "th" should be the right-hand superscript of "j". Now this is fixed (see right after equation (1)).

>>
>>The descriptions next to all the figures are too small which makes them unreadable. Please correct this.

This has been hopefully fixed by a significantly increase of the size of all the figures.

>>
>>Please describe (in Conclusions) more clearly/understandably the advantages and disadvantages of the proposed method in relation to existing ones.

Done, thank you. Please find the extended latest paragraph of section "Conclusions".

Round 2

Reviewer 1 Report

It is good for this manuscript.